# Smart joints: auto-cleaning mechanism in the legs of beetles

Konstantin Nadein 🔘 [1✉] & Stanislav Gorb 🔘 [1]

The auto-cleaning system in digging forelegs of the Congo rose chafer *Pachnoda marginata* femoro-tibial joint is described. The cleaning system consists of four subsystems: three external ones represented by microsetal pad, hairy brush and scraper and one internal one. They work proactively not only removing contaminants, but also preventing them from entering the joint. The principle of functioning of the cleaning system is based on the sliding of the contacting surfaces of the joint, equipped with hairs, bristles and scrapers. The mutual movement of such surfaces leads to the shift of contaminating particles and, ultimately, to their removal from surfaces of the joint. The key feature of the joint cleaning system is its complete autonomy, in which cleaning is performed constantly with each movement of the femoro-tibial joint without special actions required from the insect. The difference between the auto-cleaning system and self-cleaning and active grooming is also discussed.

[1] Functional Morphology and Biomechanics, Zoological Institute, Christian-Albrechts University of Kiel, Am Botanischen Garten 1-9, 24118 Kiel, Germany.
✉email: k.nadein@gmail.com

For the majority of organisms, staying clean is perhaps just as important as staying healthy. The environment continuously supplies a wide variety of organic and inorganic pollutants in all three phases: gaseous, liquid, and solid. Living organisms have evolved many mechanisms to stay clean and minimize the effects of contamination[1]. Small-sized animals, such as insects, are particularly sensitive to the external contamination, because their exoskeleton provides various functional roles, such as mechanical support, excretion, protection, feeding, sensing, and acting as a barrier against desiccation[2,3].

Mechanisms of keeping surfaces clean in insects are of two types: self-cleaning and active cleaning by grooming. Self-cleaning or so-called "Lotus effect" is a passive mechanism based on the structurally enhanced hydrophobicity (superhydrophobicity) and adhesion reduction of contaminating particles, which can be easily removed from such surfaces by the rolling water droplets[4,5]. Self-cleaning has been demonstrated for the superhydrophobic surface of cicada wings[6,7]. Similarly, nanostructured surfaces of the eyes in insects exhibit an anti-adhesive effects and corresponding self-cleaning property decreasing the real contact area between contaminating particles and the eye's surface[8]. Passive self-cleaning of the tarsal adhesive pads through repeated steps is reported for some beetles and stick-insects[9,10].

Active cleaning by grooming in insect involves the mechanical removal of contaminants and is performed with a variety of behaviorally determined movements that can be accompanied by the various specialized morphological structures[11]. As a multipurpose behavior, grooming is also executed to spread the gland secretions over the body, removing ectoparasites, to prevent bacterial and fungal fouling and to enhance olfactory acuity[12,13]. Insects are able to groom various parts of their body: antennae, head, legs, thorax, wings, and abdomen[14–17]. However, not all body parts are equally available for grooming that can be constrained, e.g., by the body shape and motility of limbs and other body joints[11].

This applies, for example, to the joints of the legs, especially their internal parts. Unlike the joints of the vertebrates, the leg joints of insects are not encapsulated and open to the environment being potentially vulnerable to the penetration of contaminants. Commonly, leg joints of insects have to work under highly contaminating conditions such as dust, sand, soil, spores, pollen, wax crystals of leaf surfaces, and often in a moist environment. Particle sizes of natural contaminants vary widely, from dozens of nanometers to millimeters[18,19]. The contamination caused by penetration of abrasive microparticles may lead to a higher friction with consequent wear and/or mechanically prevent ease and precise movements. One of the most vulnerable ecological groups of insects in this respect are probably digging insects. While burying up, the joints of their digging legs are in constant contact with soil particles whose sizes estimate in the range of 0.0002–2.0 mm[20]. In this case, preventing contaminants from entering the joint and removing particles that have already penetrated are real challenges. Self-cleaning based on the Lotus effect or reduced adhesion do not seem to be effective enough right while digging being incapable to prevent penetration of contaminants into the joint cavity, and grooming is impossible due to the inaccessibility of the joints.

In this paper, an automatically working cleaning system has been revealed in the femoro-tibial joint of digging legs of the beetle *Pachnoda marginata* (Coleoptera: Scarabaeidae) (Fig. 1a), establishing a new type of anti-contamination mechanism in insects. Using the originally developed experimental method based on the μCT technique and scanning electron microscopy, we for the first time demonstrated biomechanical principle behind such an auto-cleaning mechanism in the leg joints of beetles.

## Results

**Structure of the femoro-tibial joint.** Femoro-tibial joint of *Pachnoda marginata* consists of two counterparts, femoral and tibial, correspondingly (Fig. 1b–g and Supplementary Fig. S1a). The femoral counterpart (Fig. 1c, e, f) is situated in the distal part of the femur at its apex and is an approximately triangular cavity (seen from the side) sunken in the femur. The ventral femoral wall in the apical third is invaginated. The internal cavity of the femur is separated from the external environment by an arthrodial membrane, which connects the femur and tibia. The femoral condyle (FC) is situated apically almost at the very edge in the middle of the internal femoral wall (Fig. 1c, f). The FCs are paired toroidal structures resembling a regular, almost closed circles. Each condyle is 90–120 μm broad, the outer diameter is 450–500 μm, and the inner diameter is 240–270 μm. Most of the surface of the condyle is smooth, with no observable microstructure. Dorsally, there is the surface region called here "grater" of about 150–200 μm length covered with structures in the form of sharp, short, flat, adjacent microprotrusions oriented primarily along the circle of FC (Supplementary Fig. S1b). The sizes of the protrusions vary from 3 to 7 μm in length, the distance between the protrusions varies from 3 to 4 to less than half of the length of the protrusions. The internal surface of the femoral condyle is covered with similar microprotrusions (Supplementary Fig. S1c). The apical extreme edge of the dorsal part of the femur forms "membraneous plate" (Fig. 1c and Supplementary Fig. S1d). The plate is about 150 μm long, ca. 450 μm wide, about 35–40 μm thick at the base and 1-μm thick at the edge (Supplementary Fig. S1e). The apical edge of the femur right above the "membraneous plate" is raised in the form of a ledge covered with a row of about 30 thick and short spines (Fig. 1b, c and Supplementary Fig. S1d).

The tibial part of the joint is represented by the proximal part of the tibia immersed in the femoral cavity (Fig. 1b, e). The base of the tibia contains a large, semi-elliptical, deep depression (Fig. 1d, g). Approximately in the middle of the depression, there is a large semicircular tibial condyle (TC) (Fig. 1d, g and Supplementary Fig. S1f, g), which is inserted into the inner part of the femoral condyle and can move within it when the tibia moves (Fig. 1e and Supplementary Fig. S1a). Thus, the space between the TC and the edge of the depression forms a deep, channel-like tibial semilunar concavity (TSC) (Fig. 1d, g), which serves as a receptacle for the FC. The surface of the tibial condyle is covered with small, sharp, triangular, flat microprotrusions of 2–4 μm in length (Supplementary Fig. S1f). The distance between the microprotrusions comprises at least the length of the projection or exceeds it. The sharp tips of the protrusions are co-oriented in the same direction. The dorsal wall of the TSC is covered with a pore-bearing area serving as openings for the lubricant delivery to the contacting surfaces of the joint. Approximately in the middle of the tibial semilunar impression, there is the base of a deep, moderately narrow groove, here called the "outflow canal", directed apically relative to the length of the tibia (Fig. 1d). The bottom of the tibial semilunar concavity near the areas adjacent to the "outflow canal" is covered with sharp, flattened, adjacent microprotrusions (Fig. 1d and Supplementary Fig. S1h). Their shape varies from triangular to narrow, strongly elongated, spike-like one. Sizes range from 2–3 μm in length to about 20 μm, the spacing between them ranges from a few micrometers (between the smallest protrusions) to less than a micrometer when the protrusions are directly next to each other (between the longest protrusions). In general, the tips of the protrusions are oriented towards the "outflow canal". The extreme margin of the ventral side of TSC, called here "scraper" (Fig. 1e and Supplementary Fig. S1i, j), is sharpened and contacting closely with the surface of the femoral condyle, moving along it at the movement of the tibia.

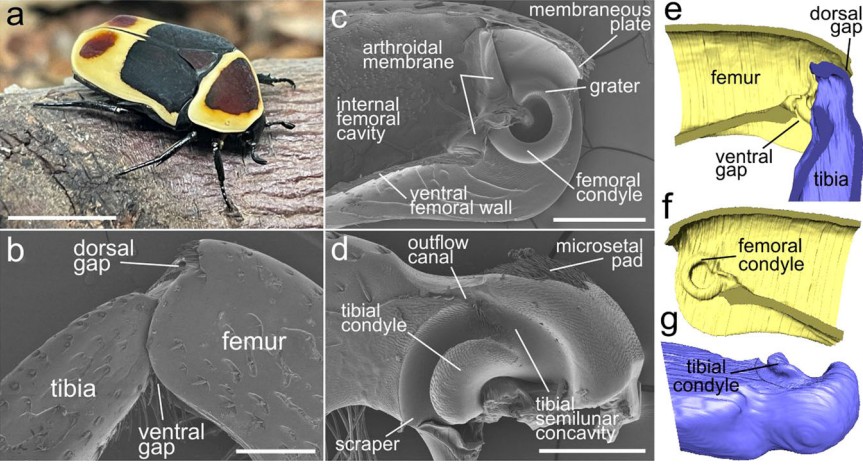

**Fig. 1 *Pachnoda marginata* beetle and the structure of the femoro-tibial joint of the foreleg. a** *P. marginata*; **b–d** SEM micrographs: **b** femoro-tibial joint, lateral view; **c** femoral counterpart, sagittal section; **d** tibial counterpart, lateral view; **e–g** micro-CT volume reconstructions: **e** femoro-tibial joint, sagittal section; **f** femoral counterpart, sagittal section; **g** tibial counterpart, anterolateral view. Scale bars: **a** 1 cm, **b**, **c** 500 μm, **d** 300 μm.

The area of the dorsal surface of the tibial base is covered in the middle with small, very dense, semi-adjacent setae that form a pad (Fig. 1d and Supplementary Fig. S1k). The setae are oriented distally. The length of this "microsetal pad" is about 250 μm, the width is about 330 μm. The length of the setae varies from 20 to 50 μm with the thickness of 2–3 μm (Supplementary Fig. S1l). The surface of the very basal part of the tibia behind the "microsetal pad" and above the TSC is covered with comparatively short, acute, flattened, adjacent microprotrusions, oriented distally.

The area at the ventral side of the tibial base is covered with some tens of long curved setae, called here "hairy brush" (Fig. 1d and Supplementary Fig. 1m). The length of setae varies from about 100–600 μm, the thickness of the setae is about 10–15 μm at the base. The length of the setae changes gradually being the shortest for the setae located more distally, whereas the longest setae are located more proximally to the tibial base. The surface of the setae is covered with small, numerous, semi-adjacent sharp notches oriented toward the apex of the seta (Supplementary Fig. S1n).

**Experimental results**. Penetration of the contaminating particles can generally happen through the gaps between the tibial and femoral counterparts of the joint. There are two major gap areas in the joint: dorsal gap and ventral gap (Fig. 1b, e).

*Microsetal pad/membraneous plate*. The dorsal gap is the space between the dorsal surface of the tibial base, where the microsetal pad is located, and the dorsal wall of the femoral apex, where the membrane plate is situated (Fig. 2). The size of the gap varies depending on the position of the tibia relative to the femur and ranges from about 30–100 μm (Supplementary Fig. S1o, p); the minimal value is observed when the membrane is situated directly above the microsetal pad. Thus, it can be assumed that the membrane plate and microsetal pad perform a protective function and prevent particles from entering the inner cavity of the joint. In order to evaluate this hypothesis, experiments with contamination by the metallic particles of various sizes have been performed (Fig. 2a–d and Supplementary Figs. S2–S4). The results of observations and measurements show that the number of particles penetrated into the joint was higher in all samples with the removed microsetal pad and membraneous plate (Fig. 2e).

*Hairy brush*. Another possible place of entry of contaminants is the ventral gap. When the tibia extends, the space between the tibial and femoral parts of the joint increases, in which the femoral part of the joint becomes more exposed, in particular its femoral condyle. When the tibia is positioned at about 90° relative to the femur, the ventral gap between them is completely covered (filled) with the hairy brush (Fig. 3). It can be assumed that the hairy brush can perform a barrier function and prevent the penetration of contaminants. This assumption was verified experimentally by applying contaminant particles to the space between the tibia and the femur (Fig. 3a, b and Supplementary Fig. S5a–d). The tibia was set in motion to the maximum possible angle for 100 times. Observation of surfaces demonstrated that after the movement of the tibia, the number of particles on the femoral surface was visually decreased and surface became cleaner (Fig. 3a, b and Supplementary Fig. S5a–d). Our analysis of the movement of the femur and setae of the hairy brush shows that at an angle of 90°, the longest proximal setae lie on the femur surface (Fig. 3c). When the tibia is flexing, the setae begin to move along the surface of the ventral outer wall of the femur (Fig. 3d). Moreover, as the angle decreases (relative to the femur), an increasing number of setae come into contact with the femur surface. They also begin to move along its surface, and their total density (with decreasing distance between them) increases.

*Scraper*. In addition to the hairy brush, the so-called scraper, based on its location, can serve as a barrier to the penetration of contaminants. When the tibia is extended, the ventral surface of the femoral condyle is exposed, being becoming accessible for the penetration of contaminants. However, when the tibia is flexing to the femur, the scraper (Fig. 3), which is very tightly adjoining the femoral condyle, slides along its surface. It can be assumed that with such a movement, it can shift the particles that have fallen on the condyle surface. This assumption was verified experimentally by applying contaminating particles to the exposed surface of the femoral condyle, followed by the movement of the tibia for 100 times (Fig. 3e, f and Supplementary Fig. S5e–j); the hairy brush was removed in this case. Examination showed that the surface of the femoral condyle became cleaner after only ten movements of the tibia (Fig. 3f). This experiment, along with other evidence, also demonstrates the action of the hairy brush as a control device. In this case, when the hairy brush is removed, it can be seen that the outer surface of the ventral wall

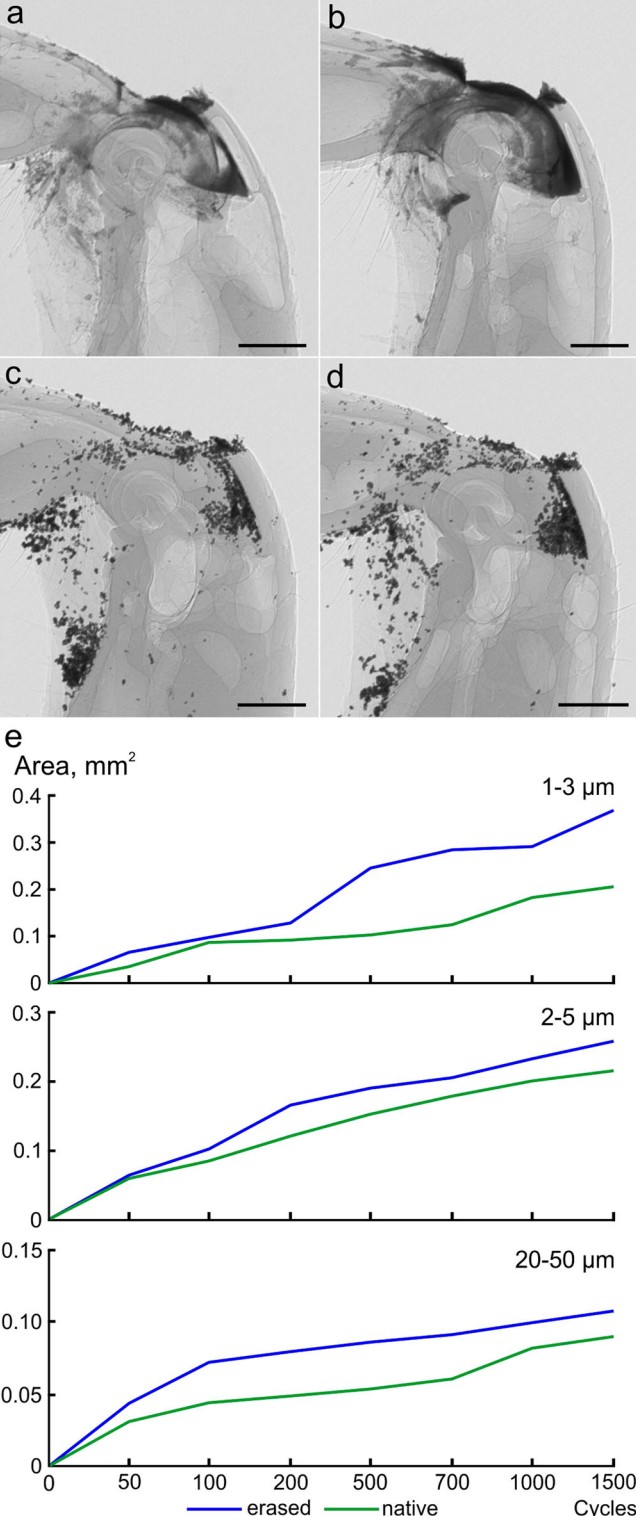

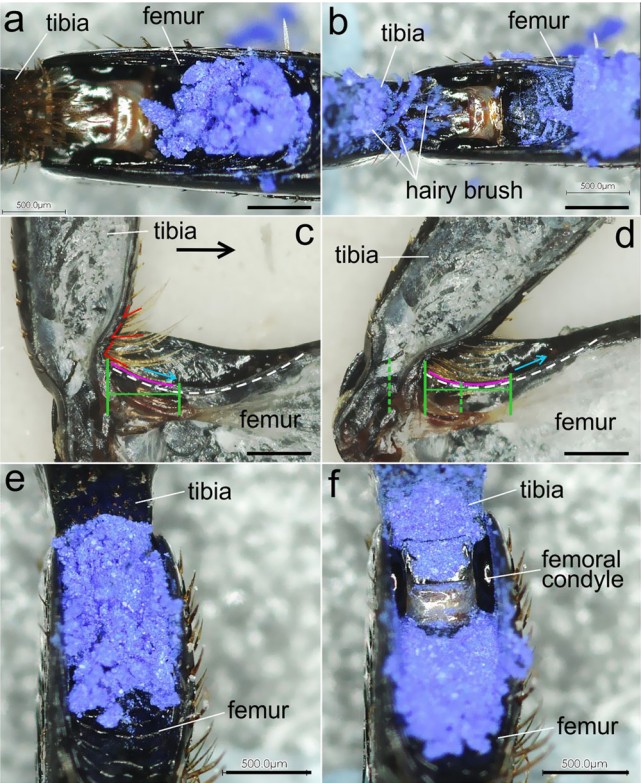

**Fig. 2 Contamination experiment with the microsetal pad and membrane plate based on the micro-CT data. a–d** Micro-CT projections of the joint: **a** femoro-tibial joint with metallic particles (1–3 µm) after 1500 opening and closing cycles, lateral view; **b** the same, with microsetal pad and membraneous plate removed; **c** femoro-tibial joint with metallic particles (20–50 µm) after 1000 cycles, lateral view; **d** the same, with microsetal pad and membraneous plate removed; **e** rate of penetration of metallic particles into the joint cavity for the particles (sizes 1–3 µm, 2–5 µm, and 20–50 µm) with removed and not removed microsetal pad and membraneous plate. The values for all particles sizes are statistically significantly different (paired $t$ test). Scale bar: **a–d** 300 µm.

**Fig. 3 Contamination experiment with hairy brush and scraper. a**, **b** Blue iron oxide powder applied to the ventral side of femoro-tibial joint external surface: **a** initial position; **b** after ten cycles; **c**, **d** movement of the tibia and corresponding movement of long hairs of the hairy brush, dotted green lines show the displacement of hairs on the femoral surface, red angles show inclination of hairs, pink line indicates the position of a hair on the femoral surface; **e**, **f** surface of the tibia and femur covered with blue iron oxide powder: **e** initial position; **f** after ten cycles. Scale bar: **a–f** 500 µm.

of the femur contains more particles (Fig. 3f) than in the experiment, where the hairy brush was not removed (Fig. 3b).

*Internal cleaning system*. As follows from the above data, contaminant particles can get inside the joint, especially those of a small size. The inner surfaces of the joint, such as the TSC, the TC, the inner surface, and part of the outer part of the FC are covered with small and dense microprotrusions. The location of microprotrusions and their orientation suggest their possible participation in the removal of microparticles trapped inside the

joint. This assumption was verified experimentally by applying metallic microparticles of different sizes to the ventral surface of the joint. The overall design of the experiment followed that of the experiment with microsetal pad (Fig. 4). The results of the experiment show that microparticles that penetrate through the ventral gap between the tibial and femoral counterparts of the joint penetrate in the TSC, on the surface of the TC, on the microstructured surface of the FC and on its inner surface (Fig. 4a, c and Supplementary Figs. S6–S8). At the same time, a number of particles was found in the outflow canal, including those bogged down in the lubricant (Fig. 4b, d). Our analysis of the micro-CT data demonstrates that the penetration of particles and their movement within the joint occurs sequentially. Starting from the point of entry into the joint, the particles move through

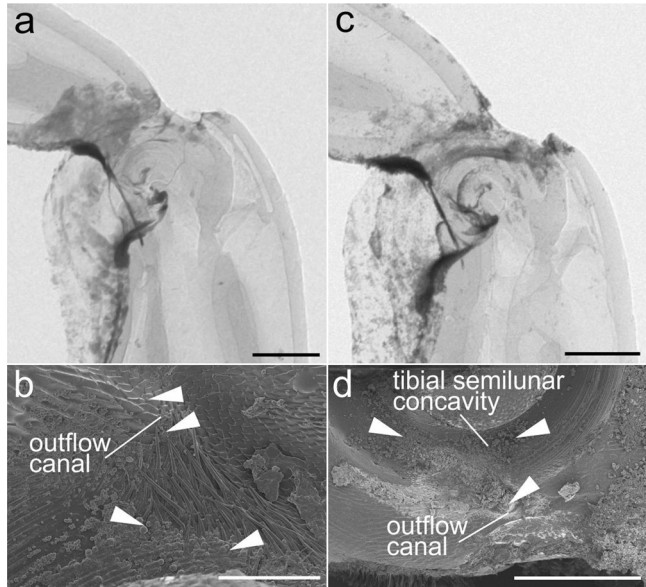

**Fig. 4 Experimental results for contamination of internal cavity of the joint based on micro-CT and SEM data. a** Femoro-tibial joint with metallic particles (1–3 μm) after 1500 opening and closing cycles, lateral view, micro-CT image; **b** the same joint sample, sagittal section, SEM micrograph, white arrowheads indicate the particles in the outflow canal and tibial semilunar concavity; **c** femoro-tibial joint with metallic particles (2–5 μm) after 1500 cycles, lateral view, micro-CT image; **d** the same joint sample, sagittal section, SEM micrograph, white arrowheads indicate the particles in the outflow canal and tibial semilunar concavity. Scale bars: **a**, **c** 300 μm, **b** 50 μm, **d** 200 μm.

the space between the TSC and the FC, some particles then penetrate to the inner surface of the FC (between it and the TC), whereas some particles continue to move within the TSC toward the outflow canal (Supplementary Figs. S6 and S7). This experiment also made it possible to evaluate the efficiency of the scraper. It looks like that only particles smaller than ca. 10 μm can penetrate into the joint in a certain amount and only in the absence of the hairy brush (Supplementary Figs. S6–S8).

For the comparison, the experiments with contamination were performed for darkling beetle *Zophobas morio* having simple walking leg without evident structures that can be attributed to cleaning mechanism[21,22]. As follows from the experimental results the leg joint of *Zophobas morio* can easily be contaminated by metallic particles of 2–5 μm (Supplementary Fig. S9).

## Discussion

The digging legs of *Pachnoda marginata*, as follows from the experimental data, are able to some extent to resist external contamination by particles (e.g., soil) with the help of specialized structures organized into a joint cleaning system (Figs. 5 and 6). The functioning of each of these structures acting in concert is discussed in more detail below.

**Microsetal pad/membraneous plate**. Experimental results indicate that the penetration of contaminants through the dorsal gap is hampered by the presence of the microsetal pad and the membraneous plate (Fig. 5a). The mechanism of functioning of this cleaning system is supposed as follows (Fig. 6a). The particles of the substrate are in contact with the microsetal pad, being located, depending on the size, both among the setae and on the surface of the pad. When extending the tibia, the microsetal pad moves directly under the membraneous plate, while the distance

between the tibia and femur in this region is minimal. As a result of the translational movement, the membraneous plate displaces (scrapes) particles from the surface of the pad, the size of which exceeds the size of the gap between the membraneous plate and microsetal pad. Since the orientation of the bristles is co-directed with the movement of tibia during its extension, the setae of the microsetal pad do not impede the movement of the membraneous plate over its surface. In addition, the bristles may reduce the contact area of the particles with the underlying cuticle surface, i.e., they minimize the possibility of stiction by adhesion. It can be assumed that as a result of this, the particles that are on the bristles and are stuck among the setae are unstable and can be easily displaced by the membraneous plate.

**Internal cleaning system**. Removal of particles that have penetrated into the internal cavity of the joint is necessary to prevent abrasive wear of the surfaces. The particle removal mechanism consists of the contacting surfaces of the joint covered with microprotrusions and works in the interaction as follows (Figs. 5b and 6b). In the internal surface of the joint, particles can penetrate into the TSC, i.e., between the concavity and the femoral condyle (Supplementary Figs. S6j–n and S7j, k), and on the inner surface of the femoral condyle (Supplementary Fig. S6l), that is, between the FC and the TC. The surface of the dorsal part of the femoral condyle (or grater) is covered with microprotrusions oriented co-directionally to the circumference of the condyle, and the surface of the TSC near the outflow canal is covered with microprotrusions oriented towards the canal. When the tibia moves, the grater on the femoral condyle moves within the TSC and slides along the corresponding microprotrusions in the concavity. The particles captured by the microprotrusions of the grater (Supplementary Fig. S6k) are displaced toward the outflow canal, when the tibia is flexing (Fig. 6b). During the reverse movement of the tibia, the particles are retained by the microprotrusions of the TSC and cannot move back to their original position. During the flexing of the tibia, the microprotrusions of the grater displace the particles even further towards the outflow canal. Since all the microprotrusions in the TSC are oriented toward the outflow canal (Fig. 5b), the particles ultimately end up in it and are removed from the inner cavity of the joint. The mechanism of removal of particles from the inner surface of the FC works in the same way. The microprotrusions of the FC and TC are co-directed, and the microprotrusions of the femoral condyle are oriented towards the grater. Particles from the inner surface of the FC are relocated by the movement of the TC (Supplementary Fig. S6n) to the grater, where they are removed in the same way as described above. SEM images of the outflow canal show the presence of microparticles trapped in the lubricant. This could presumably be evidence of one more function of the lubricant being involved in the entrapment of contaminants particles that facilitates their removal[21,22].

**Scraper**. The ventral surface of the femoral condyle, as well as the space between it and the tibia, is potentially vulnerable to the entry of contaminants. When the tibia is fully flexed, the condyle surface is completely covered by the corresponding part of the tibia and the hairy brush. When the tibia is opened, the surface of the condyle becomes accessible and particles that are not retained by the hairy brush can get on it. The penetration of such particles into the joint can lead to wear on the inner surfaces of the joint (Supplementary Fig. S6j, i, m). The direct participation of the scraper (paired ventro-lateral protrusion of the articular part of the tibia) in cleaning the surface of the femoral condyle is experimentally confirmed here. The edge of the scraper is adjacent to the surface of the condyle and the gap between

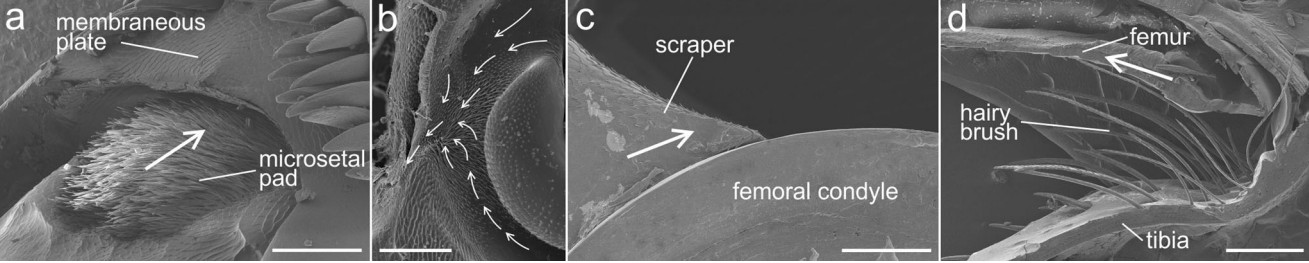

**Fig. 5 Auto-cleaning structures in the femoro-tibial joint of *Pachnoda marginata*. a** Microsetal pad/membraneous plate; **b** internal cavity; **c** scraper; **d** hairy brush. Scale bars: **a**, **b** 100 μm, **c** 50 μm, **d** 200 μm.

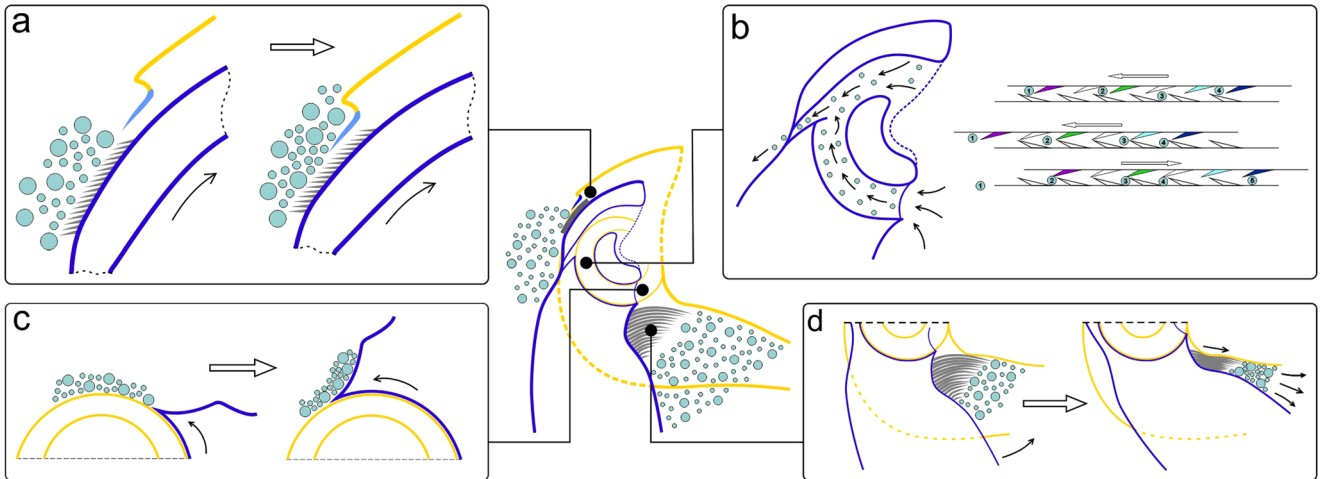

**Fig. 6 Schematic representation of the auto-cleaning mechanisms functioning. a** Microsetal pad/membraneous plate; **b** internal cavity; **c** scraper; **d** hairy brush.

them is less than 2–3 μm (Fig. 5c). When the tibia is flexing, the edge of the scraper slides over the surface of the condyle and displaces particles from its surface (Fig. 6c). As a result, when the tibia is fully flexed, a clot of displaced particles falls on the surface of the femur, where they are removed by the setae of the hairy brush according to the mechanism described below.

**Hairy brush.** The hairy brush appears to be a barrier against penetration of particles through the ventral gap. As follows from our observations, the hairs fill the space between the femur and tibia as fully as possible at an angle of no more than 90° (Fig. 5d). As shown in our experiments, flexion of the tibia to the femur results in the particle displacement (pushing out) and cleaning of the femoral surface. This is achieved due to several features. (1) The setae are curved and their curvature corresponds to the curvature of the femoral surface. (2) The length of the setae gradually decreases in the distal direction of the tibia. (3) The angle of inclination of the setae changes from approximately straight for the longer setae on the tibial base to the sharp one for the shorter setae situated more distally. Thus, the cleaning mechanism by means of the hairy brush can be described as follows (Fig. 6d). When the tibia starts moving, the longer setae situated closer to the base lie down on the femoral surface and begin to move along it. As the tibia approaches the femur, more and more setae come into contact with the femur surface. In this case, due to the corresponding angle of inclination of the setae, the latter orient parallel to the femur surface. At the same time, the density of the setae increases, and the distance between them decreases. The movement of setae along the surface of the femur leads to the displacement of particles proximally relative to the femur, that is, further from the ventral gap. This is also facilitated

by the presence of notches on the surface of setae, whose sharp tips are directed from the base to the apex (Supplementary Fig. S1n). Obviously, the role of the notches is to prevent the movement of particles between the setae. This arrangement of notches on one setal side increases and prevents the movement of particles between the setae. Also, the arrangement of notches promotes pushing the particles out, when the setae move relatively to each other. Some notches trap the particles, whereas the others (from adjacent hairs) scrape them towards the tips of the setae.

Obviously, none of the above mechanisms provide absolute protection against particle penetration. However, there are additional indirect mechanisms that may be of importance during the leg movement, when digging. As indicated above, the maximum dimensions of gaps, in which contaminants can penetrate, appear, when deviating from the optimal angle (about 80–90°). In such positions, especially, when the angle starts to increase by more than 90°, the dorsal gap increases, because the area of the ventral surface of the femoral condyle became not covered by the tibial part. With constant movement of the tibia, especially with greater amplitude, the likelihood of penetration of microparticles is supposedly increases. It could be supposed that, when digging, the beetles try to keep their tibiae in a position of 80–90° relative to the femur and move them minimally during digging. This accomplishes the goal of keeping the gaps as small as possible and minimizing particle penetration. It is also important to note that at this position of the tibia, the hairy brush maximally fills the gap between the tibia and femur. The tibial flexion to an angle greater than 90° leads to an increase of the distance between the setae of the hairy brush and the femur and to a decrease of their effectiveness as a barrier.

Thus, the auto-cleaning system of *Pachnoda marginata* leg joint is represented by the structural and functional complex of

structures, which can be subdivided into four subsystems: (1) microsetal pad/membraneous plate, (2) internal cleaning subsystem, (3) hairy brush, and (4) scraper (Figs. 5 and 6). Two of these subsystems, namely the first and the third ones, not only clean the outer parts of the joint, but also prevent the penetration of particles into the joint cavity, performing a barrier function. The mutual arrangement of surfaces, the structure and orientation of their structural surface elements result in their interaction at every movement of the joint, i.e., flexion-extension of a tibia, which in turn leads to the removal of contaminant particles to the outside of the joint. This cleaning mechanism working exclusively due to the structural and functional organization of the joint and not requiring any special actions is considered here as automatic cleaning system.

The principle of the auto-cleaning system is based on the interaction of the surfaces due to their relative motion and the corresponding structural elements, such as various microprotrusions, setae and scrapers. In this sense, there is an obvious similarity to the active cleaning through grooming. The functioning of the internal cleaning system in the joint (Fig. 6b) described here resembles previously described grooming mechanism of insects, scraping particles with angled bristles[11]. Another example is cleaning of the antennas (as well as the legs) that can be performed either by the mouthparts or by specialized antennal cleaners[11] covered with setae and bristles (by the terminology of Hlavac[11]). The latter are situated on the forelegs and bear various setae in special depressions on the tibia as for example, inground beetles Carabidae[23]. In some insect groups, such as Hymenoptera, antenna is cleaned by passing through a modified tibial spur and a notch with a comb of bristles on the first protarsomere[16].

It may be supposed that leg joint structures involved in the cleaning mechanism in *Pachnoda marginata* are the result of specialized adaptation for digging. Thus, the legs of beetles not adapted to burrowing may not have such adaptive structures, as seen by the example of the simple walking legs of darkling beetle *Zophobas morio*[21,22] whose joints can easily be contaminated (Supplementary Fig. S9).

There are only very limited options for cleaning joints through grooming, with significant limitations and low effectiveness, such as accessibility of only the external surface and only for those species, whose body structure and joint mobility allow for it. For instance, in Hymenoptera legs can be groomed by rubbing against each other and principally it cannot be excluded the possibility for these insects to clean external surface of joints in this way followed by observations reported in ref. [16], though it was not specifically emphasized. Obviously, in the case of burrowing insects, such as the model object of this study, any grooming of this kind can be excluded. Nonetheless, it is theoretically possible to groom the dorsal surface of the joint by rubbing it with the tarsus of the legs from the anterior pair, for example, to rub the joint of the hind legs by the tarsus of the middle one, although this type of grooming has never been recorded[14]. It is also potentially possible to clean the external surface of the femoro-tibial joint by rubbing against the elytra or abdomen, but this is only potentially possible from the side facing the corresponding surface. One way or another, grooming is excluded as a means to clean the inner cavity of the joint.

**Conclusions**. The auto-cleaning system found in *Pachnoda marginata* leg joints has a number of features that distinguish it from both passive self-cleaning and active grooming.

1. Behavioral pattern of movements are not involved as in grooming and there are no restrictions associated with body and limbs shape and 3D joint kinematics.

2. Cleaning is carried out by the interaction of two surfaces within one organ of the body, but not due to the interaction of different organs of the body.

3. Cleaning is performed by the mechanical removal of contaminating particles at the movement of the contacting surfaces, but not due to the anti-adhesive properties of the surface alone like in Lotus effect.

4. The cleaning system is autonomous, i.e., does not require any special actions from the insect and works constantly with every movement of the joint.

5. The cleaning system works both preventively, impeding the penetration of contaminants and actively, removing particles of contaminant that already have got inside.

The principles of functioning of the auto-cleaning system of joints in digging beetles may be of potential interest for further research in bionics/biomimetics, in particular, in areas related to legged robotics and microelectromechanical devices. The joints and hinges of such devices may be designed in a similar way to those of insects and may also be exposed to the environment. In this regard, similar mechanism of protection from contaminants may be of advantage for them.

## Methods

**Study specimens and preparation**. Beetles of Congo rose chafer *Pachnoda marginata* (Drury, 1773) (Coleoptera: Scarabaeidae) and darkling beetle *Zophobas morio* (Fabricius, 1776) (Coleoptera: Tenebrionidae) were purchased at the larval stage from commercial suppliers (MD-Terraristik, Germany) and then kept and reared in the colony at the Laboratory of Functional Morphology and Biomechanics at the Kiel University. Legs from freshly $CO_2$-anesthetised individuals were cut off and dissected with a razor blade.

**Scanning electron microscopy (SEM)**. The observations were made at an accelerating voltage of 3 kV using a Hitachi S4800 (Hitachi High-Technologies Corp., Japan) scanning electron microscope. The stubs with glued legs were air dried at room temperature for at least 24 h and then coated (thickness 10 nm) with gold-palladium using a Leica EM SCD500 sputter coater (Leica Microsystems GmbH, Wetzlar, Germany).

**X-ray micro-computed tomography (micro-CT)**. For the anatomical studies scanning of the dry foreleg was carried out by a SkyScan 1172 (Bruker Corp., Billerica, USA.) at 40 kV and 250 µA, with a camera pixel size of 8.8 µm, image pixel size 2.0 µm; 1573 projections were recorded over the 180° rotation. For 3D reconstruction, the graphic segmentation tool software Amira® 6.2 (FEI Company, Visage Imaging, Germany) has been used. For the experiments, the scanning has been done in a transfusion mode with the same parameters and without rotation.

### Experiments

*Experiments with "microsetal pad/membraneous plate"*. Two sets of the fresh samples of the forelegs were tested: one with the microsetal pada erased by scrubbing with a razor blade and with the membraneous plate cutoff by a razor blade and the other in the native state as a control. Dorsal surface of the tibial base has been covered by the metallic particles (iron) of the following sizes: 1–3 µm, 2–5 µm, and 20–50 µm. The tibia was set in motion by hand, allowing extension and flexion of 1500 times to the maximum possible opening angle (named here cycles). Every 50 times a portion of particles was applied on the dorsal surface of tibia. The penetration of particles into the joint cavity was recorded using micro-CT after 50, 100, 200, 500, 700, 1000, and 1500 movements of tibia. Metallic, electron-dense particles are clearly visible in X-rays and contrast sharply against the background of a less dense cuticle. The conditional quantity of particles that penetrated into the articular cavity was calculated as occupied dark area in the images using image analysis software SigmaScan Pro 5.0.0 (SPSS Inc., USA). The measurement data are summarized in Supplementary Table S1. Every sample after completion of the full amount of movements was dissected in sagittal plane and examined in SEM as described above.

*Experiments with the internal cleaning system of the joint*. In general, the experimental setup follows that of described above. Fresh samples of the forelegs were tested with the hairy brush removed. The metallic particles of the sizes 1–3 µm, 2–5 µm, and 10–30 µm were applied on the ventral surface of the femoro-tibial joint.

*Experiments with scraper*. A fresh sample of the foreleg has been glued on an object glass at the dorsal surface of the femur with the tibia remained movable. The hairs

of hairy brush were removed. Tibia was extended, and blue iron oxide powder has been applied to the surface of the joint. The tibia was set in motion, allowing extension and flexion of 100 times to the maximum possible angle. The surface has been photographed by a 3D measurement microscope Keyence VR 3000 (Keyence Corp., Japan) after 10, 20, 40, 80, and 100 movements.

*Experiments with the hairy brush.* The experimental setup follows the one described above for the experiments with the scraper with the only difference that the hairs of the hairy brush were not removed. The surface has been photographed by a 3D measurement microscope after 10, 50, and 100 movements.

*Experiments with Zophobas morio.* Fresh samples of the forelegs were tested by contamination with 2–5 μm metallic particles of the dorsal gap and ventral gap. The experiment setup follows those described for *Pachnoda marginata*.

**Statistics and reproducibility**. The data have been statistically analyzed using SigmaPlot 12.5 (Systat Software Inc., USA). Based on the paired $t$ test, the values for all particle sizes for removed microsetal pad and membraneous plate were statistically significantly different in comparison with control (for 1–3 μm: $t = 3.752$, DOF = 6, $P = 0.009$; for 2–5 μm: $t = 5.159$, DOF = 6, $P = 0.002$; for 20–50 μm: $t = 8.151$, DOF = 6, $P = < 0.001$).

**Reporting summary**. Further information on research design is available in the Nature Research Reporting Summary linked to this article.

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

## Acknowledgements
The work was funded by DFG research grant NA 12647/2-1 "Functional design of beetle leg joints: morphology, tribology, and cuticular microstructure" to K.N. and by DFG research grant GO 995/38-1 to S.G.

## Author contributions
K.N. and S.G. developed the scientific question and prepared the study design. K.N. carried out the experiments, calculations, and observations and prepared the manuscript and figures. All the co-authors discussed the results and revised the manuscript.

## Funding

## Competing interests
The authors declare no competing interests.
