## [Peer Review File · Communications Biology]

Reviewers' comments:

Reviewer #1 (Remarks to the Author):

In the present study, Nadein & Gorb describe the foreleg femorotibial articulation of the Congo rose chafer, *Pachnoda marginata*, which is a scarab beetle that digs in sand or soil as part of its natural lifecycle. The authors observed a (highly derived) conformation of this articulation that involves very tight structural fit of the component leg segments. Raising the question of how such a fine mechanical system can avoid wear during the process of digging, and how a joint can be cleaned without direct grooming by other body parts, the authors evaluated the morphology of the joint using μ -CT and SEM technologies. The authors observed four elements that may have a role in the autonomous cleaning of the joint: (1) a pad of microscopic setae and a "membranous plate" that may act akin to the strigil of Hymenoptera (particle capture by hairs then scraping); (2) a groove present in both the anterior and posterior primary articulations that is not associated with inter-part motion (i.e., a potential outflow gutter); (3) a dense brush of setae on the proximoventral surface of the tibia (a possible physical barrier to grains); and (4) a sharp, wedge-shaped outgrowth of the proximoventral tibial apex that resembles a scraper. To test these functional hypotheses, the authors isolated the legs, covered them in metallic beads, then moved the joints 100 to 1000 times, using both qualitative interpretation of light microscopy and μ -CT to quantify beads in the joint after X number of flexions and extensions, with or without experimental manipulation. The authors found that removal of the "membranous plate" and the barrier-like hair brush had a statistically significant effect on bead count, and that each of the enumerated elements contributed to reduction of beads in the soft tissues of the joint.

Overall, the study is of high biomimetic interest, and the results of the present study are as convincing as they could be without comparison to species that do not have such a modified foreleg femorotibial articulation. Indeed, the only meaningful shortcoming of the work is that the analysis is conducted on the joint as if it were an "ideal" system, outside of the evolutionary context which has shaped the articulation. The work would be improved if the authors were able to demonstrate the derived nature of the articulation. The simplest approach would be through literature citations in the introduction. Alternatively, the work could be substantially improved if the authors were to replicate the experiment with an insect that has the plesiomorphic or "generalized" condition of the leg. Because this would be an undertaking of equivalent effort as the work in its current form, I leave it to the authors and editor(s) to decide if this were to be necessary. However, if such a test were done, this would be convincing beyond a doubt; at present, the work is strongly indicative and stands as a reasonable evaluation of the hypothesized autocleaning mechanism. To reiterate the logic: (1) The mechanism is derived (but this isn't clearly demonstrated by the authors, although it is obviously true from the Principles of Insect Morphology [Snodgrass, 1935]) and (2) the observations of this mechanism are made quantitatively and experimentally, but (3) there is no final contrast to a simpler, plesiomorphic condition, thus one could still assume that this system is not evolutionarily adapted "for" autocleaning, or does not represent an improvement on the system. From the mechanical perspective this could be adequate; from the evolutionary, there is one last comparison that is desired. Below I have provided more specific comments, all of which are minor and relate to word choice or text clarity. In this regard, the text would be improved if it were written in a direct, rather than telegraphic style, i.e., by including the noun articles as often as possible. Note that this latter point is extremely minor and just a suggestion. Finally, to ease the processes of peer review and text revision, please provide line numbers.

Structure of the femorotibial joint.

- "Immersed" does not sense in the context of the following line, as it implies that the femoral apex is sunken into a liquid: "The femoral counterpart (Fig. 1c, e, f) is situated in the distal part of the femur at its apex and is a conditionally triangular cavity (seen from the side) immersed in the femur." I think that in both instances of the use of "immersed" the word "sunken" would be an appropriate substitution.

- Also, for that same sentence, please clarify what "conditionally triangular cavity" means. Do the authors mean that the shape of the cavity varies? If so, what other shapes does the cavity take? (And when is it actually triangular?) If the shape of the cavity does not vary, then I cannot understand what is meant by "conditional".

- The suggested term "membrane plate" does not make sense on its own ("[t]he apical edge of the

femur right above the 'membrane plate' is raised in the form of a ledge"). Please specify how presumably unsclerotized membrane forms a "plate", as this implies that the membrane is actually in the form of a sclerite.

- Regarding the "grater", does this structure have a formally recognized name in other Coleoptera, or is this a novel observation? In general, clarifying which elements of this articulation are new observations would improve the value of the present work.

- "Setae" misspelled on p. 4 ¶2 line 2.

- P. 4 ¶3: "... the thicknesses of the setae are about ..."; "[t]he surfaces of the setae are covered with ..."

Experiments.

- How were the microsetal and membranous pads "erased"? Erasure implies a scrubbing action. Were these structures scrubbed off?

Experimental results.

- Microsetal pad/membranous plate.

o ¶1. I recommend using "evaluate" or "test" rather than "prove" in the phrase "in order to prove this hypothesis".

- Scraper.

o Suggested text revision (grammar): "... the so-called scraper, based on its location, can serve as a ..."

o Suggested text revision: "This experiment, along with other evidence, also demonstrates ..."

Discussion.

- "Thus, the cleaning mechanism by the means of the hairy brush ..."

- "... the likelihood of penetration of microparticles is supposedly increases."

- "... and move them minimally during digging."

- By the terminology of which author?—"... (by the terminology of author)."

Reviewer #2 (Remarks to the Author):

Nadein and Gorb's MS deals with the investigation of the leg cleaning system of a burrowing beetle. By implementing classical electron microscopy (SEM) and micro-CT techniques, the authors have morphologically characterized the structures involved in the cleaning mechanisms of the tibia-femur joint and articulation point. Furthermore, through a series of experiments involving contamination with metal particles of different sizes, the authors highlighted the self-cleaning mechanism that ensures the prevention (or at least reduces the risk) of particle entry. The MS is well written throughout, the scientific approach is of note, the experimental design is excellent, and so is the presentation and discussion of the results. Therefore, the MS can be accepted in its present form.

Only one annotation concerns the technique used to set in motion the leg after covering with metallic particles (Pag. 13 top). Authors should specify whether this movement was created using specific equipment or "by hand".

Response to Reviewers

Reviewer #1 (Remarks to the Author):

Response. Thank you very much for the thorough peer-review and valuable suggestions. The experiments with contamination of the leg joint of darkling beetle *Zophobas morio* is performed. This species has simple walking legs without evident structural adaptations for cleaning and can be considered as plesiomorphic condition. The corresponding text and illustrations are added to the manuscript.

Structure of the femorotibial joint.

- “Immersed” does not sense in the context of the following line, as it implies that the femoral apex is sunken into a liquid: “The femoral counterpart (Fig. 1c, e, f) is situated in the distal part of the femur at its apex and is a conditionally triangular cavity (seen from the side) immersed in the femur.” I think that in both instances of the use of “immersed” the word “sunken” would be an appropriate substitution.

Response. Changed to ‘sunken’.

- Also, for that same sentence, please clarify what “conditionally triangular cavity” means. Do the authors mean that the shape of the cavity varies? If so, what other shapes does the cavity take? (And when is it actually triangular?) If the shape of the cavity does not vary, then I cannot understand what is meant by “conditional”.

Response. It means approximately of triangular shape. The corresponding correction is done.

- The suggested term “membrane plate” does not make sense on its own (“[t]he apical edge of the femur right above the ‘membrane plate’ is raised in the form of a ledge”). Please specify how presumably unsclerotized membrane forms a “plate”, as this implies that the membrane is actually in the form of a sclerite.

Response. The “membrane plate” is changed to “membraneous plate” as throughout the text.

- Regarding the “grater”, does this structure have a formally recognized name in other Coleoptera, or is this a novel observation? In general, clarifying which elements of this articulation are new observations would improve the value of the present work.

Response. This is a novel observation and thus it does not has a special term. The structure of *Pachnoda marginata* leg joint has been partially studied in a few papers. Since the leg joints of beetles (and insects) are still poorly studied morphologically the terminology is not yet stable and in its infancy. Extensive comparative morphological studies and corresponding terminology are out of the direct scope of this paper devoted to biomechanics of a single species and should be addressed to a paper devoted to comparative morphology of beetles’ leg joints.

- “Setae” misspelled on p. 4 ¶2 line 2.

Response. Corrected.

- P. 4 ¶3: “... the thicknesses of the setae are about ...”; “[t]he surfaces of the setae are covered with ...”

Response. Corrected.

Experiments.

- How were the microsetal and membranous pads “erased”? Erasure implies a scrubbing action. Were these structures scrubbed off?

Response. The microsetal pad has been erased by scrubbing with a razor blade and membranous pads were cut off by a razor blade. The corresponding text is added.

Experimental results.

- Microsetal pad/membranous plate.

o ¶1. I recommend using “evaluate” or “test” rather than “prove” in the phrase “in order to prove this hypothesis”.

- Scraper.

- o Suggested text revision (grammar): "... the so-called scraper, based on its location, can serve as a ..."
- o Suggested text revision: "This experiment, along with other evidence, also demonstrates ..."

Response. Corrected.

Discussion.

- "Thus, the cleaning mechanism by the means of the hairy brush ..."

Response. Corrected.

- "... the likelihood of penetration of microparticles is supposedly increases."

- "... and move them minimally during digging."

Response. Unclear what changes are required since it repeats what is written in the manuscript.

- By the terminology of which author?—"... (by the terminology of author)."

Response. By the terminology of Hlavac, 1975. The corresponding text is added.

Reviewer #2 (Remarks to the Author):

Nadein and Gorb's MS deals with the investigation of the leg cleaning system of a burrowing beetle. By implementing classical electron microscopy (SEM) and micro-CT techniques, the authors have morphologically characterized the structures involved in the cleaning mechanisms of the tibia-femur joint and articulation point. Furthermore, through a series of experiments involving contamination with metal particles of different sizes, the authors highlighted the self-cleaning mechanism that ensures the prevention (or at least reduces the risk) of particle entry. The MS is well written throughout, the scientific approach is of note, the experimental design is excellent, and so is the presentation and discussion of the results. Therefore, the MS can be accepted in its present form.

Only one annotation concerns the technique used to set in motion the leg after covering with metallic particles (Pag. 13 top). Authors should specify whether this movement was created using specific equipment or "by hand".

Response. Thank you very much. The leg joint parts (tibia) were set in motion by hand. The corresponding text is added.

REVIEWERS' COMMENTS:

Reviewer #1 (Remarks to the Author):

After considering the review response and the revised manuscript, I think that no further revisions are strictly required. Some more effort could be made to provide valuable context the unusual conformation of the joint in *Pachnoda*, but this is not necessary. The addition of the "generalized" *Zophobas* is useful, and could be used to more clearly make the case that the system in *Pachnoda* is special. Another point is that, barring an extensive survey of coleopteran femorotibial joints, it is possible that the form of *Pachnoda* is not "unique", although the system does appear so. The final room for improvement is directly comparing *Zophobas* and *Pachnoda* more quantitatively, although this is not necessary.